# Epidemiological Characterization of Respiratory Pathogens Using the Multiplex PCR FilmArray™ Respiratory Panel

**DOI:** 10.3390/diagnostics14070734

**Published:** 2024-03-29

**Authors:** Young Jun Hong, Bo Kyeung Jung, Jae Kyung Kim

**Affiliations:** 1Department of Biomedical Laboratory Science, College of Health Sciences, Dankook University, Cheonan 31116, Republic of Korea; koko5542@naver.com; 2Department of Laboratory Medicine, College of Medicine, Dankook University, Cheonan 31116, Republic of Korea; lovegodmother@hanmail.net

**Keywords:** diagnostic test, FilmArray™ RP, multiplex PCR, upper respiratory tract infections, virus diseases

## Abstract

Various pathogens can cause upper respiratory tract infections, presenting challenges in accurate diagnosis due to similar symptomatology. Therefore, rapid and precise diagnostic tests are crucial for effective treatment planning. Traditional culture-based methods for diagnosis are limited by their reliance on skilled personnel and lengthy processing times. In contrast, multiplex polymerase chain reaction (PCR) techniques offer enhanced accuracy and speed in identifying respiratory pathogens. In this study, we aimed to assess the efficacy of the FilmArray™ Respiratory Panel (RP), a multiplex PCR test capable of simultaneously screening 20 pathogens. This retrospective analysis was conducted at Dankook University Hospital, South Korea, between January 2018 and December 2022. Samples from patients with upper respiratory tract infections were analyzed. Results revealed adenovirus as the most prevalent pathogen (18.9%), followed by influenza virus A (16.5%), among others. Notably, a 22.5% co-infection rate was observed. The FilmArray™ RP method successfully identified 20 pathogens within 2 h, facilitating prompt treatment decisions and mitigating unnecessary antibiotic prescriptions. This study underscores the utility of multiplex PCR in respiratory pathogen identification, offering valuable insights for epidemiological surveillance and diagnosis.

## 1. Introduction

Upper respiratory tract infections, commonly associated with symptoms such as sore throat, fever, runny nose, and cough, are prevalent worldwide, necessitating medical intervention for a significant number of affected individuals. The escalating need for healthcare services has led to substantial financial implications due to the high medical costs incurred [1,2]. Additionally, these infections significantly contribute to the global disease burden, with an estimated 3.5 million deaths worldwide in 2008 attributable to upper respiratory tract infections [3,4]. The pervasive nature of these infections underscores the urgency of addressing their impact on public health systems and implementing effective prevention and management strategies. Upper respiratory tract infections, caused by various pathogens such as bacteria and viruses, present a diagnostic challenge owing to their overlapping symptoms. This similarity often leads to indiscriminate antibiotic prescriptions, fostering the emergence of antibiotic-resistant strains [5]. Distinguishing between bacterial and viral infections is crucial for effective treatment. However, relying solely on symptoms for diagnosis may not suffice. Implementing diagnostic tools and guidelines can aid in accurate identification, curbing unnecessary antibiotic use and combating antimicrobial resistance. Research indicates that a significant portion of pediatric patients (approximately 21%) receive antibiotic prescriptions; of these, 44% patients present with upper respiratory tract infections, indicating a pervasive issue of antibiotic over-prescription in healthcare [6,7]. Addressing this concern requires concerted efforts to minimize unnecessary antibiotic use and adopt treatment strategies guided by precise diagnostic evaluations. Implementing accurate diagnostic tests can assist in discerning the need for antibiotics, thus promoting judicious prescribing practices and mitigating the development of antibiotic resistance.

Several diagnostic tests are available for the identification of upper respiratory pathogens, each with their own set of advantages and limitations. Antigen tests, for instance, offer rapid results but are plagued by low sensitivity [8,9]. On the other hand, traditional methods such as conventional virus culture boast higher sensitivity but entail longer processing times, potentially delaying treatment decisions [10]. Multiplex polymerase chain reaction (PCR) tests have emerged as a valuable tool in hospitals for rapidly and accurately identifying respiratory pathogens. Among these, the FilmArray™ Respiratory Panel (RP) stands out. It is an FDA-approved diagnostic tool and is capable of detecting over 20 respiratory pathogens within 2 h. However, this method requires specialized facilities and skilled personnel for testing and result interpretation. Despite its efficacy, limited studies have evaluated its utility in identifying respiratory pathogens [11]. We employed the FilmArray™ RP to epidemiologically characterize upper respiratory tract infections, assessing its suitability for such research endeavors. Our investigation aimed to furnish foundational data crucial for informing healthcare policies, including the identification of infection patterns, formulation of vaccination strategies, and containment of pathogen spread within communities. Spanning from 2018 to 2022, our study period encompassed both pre- and post-outbreak phases of the coronavirus disease 2019 (COVID-19) pandemic precipitated by severe acute respiratory syndrome coronavirus 2 (SARS-CoV-2). As such, our study could offer invaluable insights into the evolving landscape of upper respiratory tract infections amidst the COVID-19 crisis. By shedding light on the impact of the pandemic on respiratory infection patterns, our study could not only enrich our understanding of disease dynamics but also provide a framework for devising proactive measures to mitigate future outbreaks. In essence, our findings could serve as a cornerstone for evidence-based healthcare strategies aimed at safeguarding public health in the face of infectious disease threats.

## 2. Materials and Methods

### 2.1. Samples

Nasopharyngeal swab (NPS) samples, totaling 300 μL in volume, were collected for infection testing at Dankook University (Cheonan, Republic of Korea) over a period spanning from 1 January 2018 to 31 December 2022. These samples underwent analysis without centrifugation. Specifically, only refrigerated and frozen samples adhering to the conditions outlined in the manufacturer’s manual were included in this study.

### 2.2. Pathogen Identification with the FilmArray™ RP

The NPS samples underwent rigorous testing in accordance with the manufacturer’s manual for the FilmArray™ RP (BioFire Diagnostics, Salt Lake City, UT, USA). Ensuring adherence to stringent safety protocols, all tests were meticulously conducted within a biosafety cabinet, with operators equipped with appropriate personal protective equipment. The sample processing commenced with the injection of samples into the kit using a hydration solution and sample buffer, followed by insertion into the designated equipment. The testing system followed a sequential workflow, beginning with nucleic acid extraction facilitated by sample buffer and zirconium beads, followed by reverse transcription, auto-nested multiplex PCR, and concluding with melting curve analysis. Notably, the initial PCR phase involved a highly multiplexed PCR, followed by an individual PCR step incorporating a cyanine dye for enhanced sensitivity and specificity. The FilmArray™ 2.1 software was used for analyzing the DNA melting curve for every well within the PCR2 array. Upon observing the presence of PCR products in a well and confirming the melting profile aligning with PCR products, the software was used to calculate the melting temperature (Tm) of the curve. If the calculated Tm fell within the specified range designated for analysis, the result was deemed “Detected”. Conversely, if the software determined that the curve did not align within the predetermined melting range, the outcome was labeled as “Not Detected”.

The FilmArray™ RP boasts the capability to detect a comprehensive range of pathogens, encompassing four bacteria and 19 viruses. Among the bacterial targets are *Bordetella pertussis* (detection of ptxP), *Bordetella parapertussis*, *Chlamydia pneumoniae* (previously named *Chlamydophila pneumoniae*), and *Mycoplasma pneumoniae*. Additionally, the panel includes detection for various viruses such as adenovirus (AdV), coronavirus 229E (CoV-229E), coronavirus HKU1 (CoV-HKU1), coronavirus NL63 (CoV-NL63), coronavirus OC43 (CoV-OC43), human metapneumovirus (hMPV), human rhinovirus/enterovirus (HRV/EV), influenza virus A (FluA), influenza virus A H1 (FluA H1), influenza virus A H1-2009 (FluA H1-2009), influenza virus A H3 (FluA H3), influenza virus B (FluB), parainfluenza virus 1 (PIV1), parainfluenza virus 2 (PIV2), parainfluenza virus 3 (PIV3), parainfluenza virus 4 (PIV4), respiratory syncytial virus (RSV), Middle East respiratory syndrome coronavirus (MERS-CoV), and SARS-CoV-2.

In our testing process, we screened for 16 viruses (AdV, CoV-229E, CoV-HKU1, CoV-NL63, CoV-OC43, hMPV, HRV/EV, FluA, FluB, PIV1, PIV2, PIV3, PIV4, RSV, MERS-CoV, SARS-CoV-2) and four bacteria (*Bordetella pertussis*, *Bordetella parapertussis*, *Chlamydia pneumoniae*, and *Mycoplasma pneumoniae*) within the NPS samples. Each testing pouch was equipped with two positive controls, ensuring the reliability and accuracy of our results. The first positive control, known as the RNA Process Control, specifically targeted the transcription of RNA from the yeast *Schizosaccharomyces pombe*. A positive outcome from this control indicated the successful execution of all steps performed within the pouch. Similarly, the second positive control detected a dried DNA target present in the well of the array, along with the corresponding primers. A positive result from this control confirmed the successful amplification of PCR2. To validate the overall test results, both control outcomes were required to be positive. In the event that either control failed, the test was repeated using a new kit to maintain the integrity and accuracy of the testing process.

## 3. Results

A total of 6367 respiratory samples underwent analysis, revealing 1538 positive cases. Among these positive cases, 1744 pathogens were identified; these included 1351 instances of single infection, 169 double infections, 17 triple infections, and a solitary case of quadruple infection (Table 1). Among the 20 pathogens detectable using the FilmArray™ RP, 15 were identified in this study, exclusively comprising viruses (AdV, CoV-229E, CoV-HKU1, CoV-NL63, CoV-OC43, FluA, FluB, hMPV, HRV/EV, PIV1, PIV2, PIV3, PIV4 RSV, and SARS-CoV-2). The prevalence of pathogens in this study was in the order of AdV (18.9%), FluA (16.5%), PIV3 (12.3%), HRV/EV (10.4%), and hMPV (9.7%); these collectively constituted over half of the total pathogens detected. Especially, data for SARS-CoV-2 was included starting from 1 October 2022. Out of a total of 73 samples tested since the inception of the study, 28 tested positive. Among these positive samples, twenty-two exhibited a single infection, while five samples showed double infection and one sample displayed triple infection (Table 2).

Notably, the 0–10 age group exhibited the highest distribution of pathogens, accounting for 72.3% of the total age group positivity rate. AdV emerged as the most frequently detected pathogen within the 0–10 age group, comprising 17.0% of cases. Furthermore, within the 0–10 age group, AdV, PIV3, HRV/EV, and RSV constituted 90.3%, 83.6%, 95%, and 97.9% of total pathogens identified, respectively, signifying elevated positivity rates among children. Conversely, unlike other pathogens, FluA and FluB exhibited higher positivity rates in the 51–90 age group, representing 56.4% and 53.3% of the distribution, respectively (Figure 1). This variance in age group distribution underscores the differential susceptibility and manifestation of respiratory pathogens across various age demographics, highlighting the need for age-specific preventive and management strategies.

The highest number of pathogens was identified in December, representing 18.2% of all detected pathogens. Notably, a significant portion, accounting for 44.1% of the pathogens, was detected between December and February. During December and February, FluA and FluB exhibited high positivity rates of 87.2% and 71.0%, respectively. Conversely, the positivity rate of PIV3 was notably lower at 51.4% from March to May. Throughout the year, pathogens such as AdV, hMPV, PIV1, PIV3, and HRV/EV showed relatively even distribution patterns (Figure 2).

A total of 187 cases of co-infection were identified, with 393 out of the total 1744 pathogens showing a co-infection rate of 22.5%. Among the pathogens, CoV-HKU1 exhibited the highest co-infection rate at 60% (three out of five), followed by PIV4 at 55.6% (five out of nine), CoV-229E at 40% (sixteen out of forty), and CoV-NL63 at 38.2% (twenty-six out of sixty-eight). Conversely, FluB displayed the lowest co-infection rate at 8.8%, followed by FluA at 13.1% and PIV3 at 13.6%. Regarding double infections by pathogen type, AdV and HRV/EV, as well as HRV/EV and RSV double infections, accounted for the highest proportion at 8.3% (14 out of 169). Co-infections involving AdV, HRV/EV, and PIV3 represented the largest proportion of triple co-infections at 17.6% (three out of seventeen). Additionally, one case of quadruple infection was noted, involving AdV, CoV-NL63, CoV-OC43, and HRV/EV (Table 3).

## 4. Discussion

Acute respiratory infection is a prevalent disease, accounting for approximately 20–40% of outpatient cases and 12–35% of inpatient cases in general hospitals. Among these, upper respiratory infections, including nasopharyngitis, pharyngitis, and tonsillitis, account for 87.5% of all respiratory infections [12]. Both adults and children show a higher infection rate with viral respiratory pathogens than with bacterial respiratory pathogens; however, children account for more than 80% of all respiratory infection cases [13,14]. Consistent with previous findings, all 1744 (100%) of the FilmArray™ RP-positive pathogens detected in this study were viruses. With respect to the distribution of the pathogens, we found that AdV had the highest positivity rate (18.9%), followed by FluA (16.5%) and PIV3 (12.3%). Previous studies focusing on viral upper respiratory infections have reported the highest positivity rates for HRV/EV and RSV and the lowest positivity rate for FluA [15,16]. However, these results may vary depending on regional and climatic factors, necessitating further research.

In our analysis, we observed 187 instances of co-infection, leading to the detection of 393 pathogens. Among these cases, one hundred and sixty-nine involved double infections, seventeen involved triple infections, and one involved a quadruple infection, reflecting an overall co-infection rate of 22.5%. Comparable co-infection rates have been reported in previous studies utilizing the BioFire FilmArray™ RV and QIAstat-Dx Respiratory Panels, which also documented rates of approximately 20%, mirroring the findings of our investigation [17,18]. However, this rate notably exceeded that reported in a prior study using the FilmArray™ Meningitis/Encephalitis panel [19]. Nevertheless, it remained lower than the rate reported in another study employing the FilmArray™ GI panel [20].

In cases of co-infection, AdV (*n* = 93) emerged as the most frequently observed pathogen, followed by HRV/EV (*n* = 57) and FluA (*n* = 38). Particularly noteworthy were the high co-infection rates of CoV-HKU1 and PIV4, which stood at 60% and 55.6%, respectively, indicating a propensity for co-infection with other pathogens. Comparatively, in previous studies on respiratory viral infections, HRV/EV surfaced as the most common infectious pathogen, trailed by FluB and AdV [21]. Consistent with our findings, another earlier study also reported multiple co-infecting pathogens, primarily HRV/EV, followed by RSV and AdV. Furthermore, elevated co-infection rates were noted for CoV-HKU1 and PIV4 [22].

In the above study, seventeen samples exhibited triple infections, while one sample exhibited quadruple infections, with none exceeding quadruple infections. Samples with triple infections or higher accounted for 1.0% of the total positive samples. In a prior study on a related topic, triple or higher duplicate infections were observed at a rate of approximately 1.3%, which was comparable to the triple double infection rate in this study [23]. Among the eighteen samples with triple infections or higher, AdV was detected in fifteen cases (83.3%), HRV/EVs in ten cases (55.6%), Cov-NL63 in seven cases (38.9%), CoV-OC43 in six cases (33.3%), and PIV-3 in six cases (33.3%), in that order. Notably, one out of five positive samples (20.0%) for Cov-HKU1 and seven out of sixty-eight positive samples (10.3%) for CoV-NL63 exhibited triple or more duplicate infections. Furthermore, SARS-CoV-2 was detected in samples displaying duplicate infections, such as AdV and RSV.

The presence of multiple respiratory Infections carries diverse clinical implications, including prolonged pathogenic infections, extended hospitalization durations, and heightened severity of respiratory-related illnesses [24,25]. However, conflicting findings have also been reported, suggesting that respiratory co-infections may not significantly influence the prevalence or severity of the disease [26,27]. Consequently, the correlation between multiple respiratory pathogen infections and the prevalence and severity of the disease remains unclear, underscoring the necessity for comprehensive investigations into co-infections involving various respiratory pathogens.

In this study, a substantial portion of positive pathogens (1121 cases, constituting 64.3%) were detected in the 1–5 age group, underscoring notably high positivity rates among younger patients. Specifically, within the 1–5 age group, pathogens such as AdV, PIV3, RSV, and HRV/EV displayed positivity rates of 79.3%, 79.4%, 96.9%, and 81.8%, respectively, indicative of robust detection rates in this demographic. Conversely, among older adults (>49 years), FluA and FluB exhibited positivity rates of 58.1% and 53.3%, respectively, suggesting heightened detection rates in this age group. Our findings align with those of previous studies highlighting elevated positivity rates among young patients (aged < 5 years) and robust detection rates for AdV, PIV3, RSV, and HRV/EV [17,28]. However, in contrast to our observations, these studies noted low detection rates for FluA and FluB among older patients but relatively elevated rates among younger individuals.

In this study, 769 pathogens were detected between December and February, corresponding to a detection rate of 44.1%. By contrast, only 245 pathogens were detected between June and August, with a detection rate of 14.0%. In the summer of June-August, 230 samples (16.0%) out of a total of 1443 samples were positive of which AdV was the most common, with 78 (33.9%). This was followed by PIV3 with 69 (30.0%) and HRV/EV with 41 (17.8%). In the winter of December–February, 657 samples (34.5%) out of a total of 1906 samples were positive of which FluA was the most common, with 252 (38.4%). After that, 90 AdV (13.7%) and 76 FluB (11.6%) were different from the summer. These findings indicated a higher detection rate during winter than during summer. Similarly, previous studies reported higher respiratory pathogen detection rates in summer than in winter [29,30]. Research on the age and timing of respiratory pathogen infections can provide fundamental data for implementing various health policies. In particular, such data can help to determine the appropriate age and timing of vaccinations as well as aid in the understanding of the epidemiology of respiratory viruses. Therefore, continuous research and monitoring are necessary to utilize this information as a basis for public health policies.

This present study had several limitations. Firstly, as it was conducted using samples solely obtained from a single university hospital in Cheonan rather than from a range of institutions, the consideration of climatic and regional characteristics may be constrained. Additionally, since the majority of patients visiting the hospital for examinations are residents of the local community, there might be limitations in generalizing the findings to reflect universal trends in respiratory infections. In addition, clinical information related to disease severity, symptoms, and diagnosis was not included, making it challenging to assess treatment outcomes and prognosis. Among the 16 viruses and four bacterial species that could be detected using the FilmArray™ RP, MERS-CoV was not detected. Despite these limitations, in this study, we analyzed 6367 samples according to pathogens, age groups, time periods, and co-infections, allowing us to understand the patterns of respiratory pathogen infections. In particular, analyzing respiratory infections according to age and time can provide valuable clinical data for understanding epidemiological patterns and establishing vaccination strategies. Moreover, the analysis of co-infection rates by specific pathogens can help in the identification of common co-infection pathogens, which can aid in treatment planning and the administration of medications.

Traditionally, microbial culture methods serve as the foundational tests for identifying pathogens responsible for infectious diseases. It is recognized that bacterial and viral culture tests demand specialized facilities and highly trained personnel, more so than molecular biological tests. Bacterial culture typically spans from 24 h to 5 days [31]. The confirmation process for the culture of respiratory viral pathogens, which is the primary focus of this study, can extend over several days to several weeks [32]. In contrast, the FilmArray™ RP offers a rapid and efficient alternative, detecting respiratory pathogens within 2 h. This swift turnaround time facilitates timely decision-making regarding treatment. The expedited results provided by the FilmArray™ RP allow for a shift from empirical antibiotics, such as amoxicillin, clavulanic acid, macrolides, and doxycycline, to more targeted antibiotic prescriptions. This transition enables more precise and effective administration of antibiotics [33]. Furthermore, the ability to test for 20 pathogens in a single sample streamlines the generation and utilization of regional, age-specific, and seasonal data for respiratory pathogen surveillance, vaccination planning, and identification of high-risk populations across different communities. The significance of rapid and accurate diagnostic tests has gained increasing recognition, prompting the adoption of various multiplex PCR diagnostic methods, including the FilmArray™ RP [34]. Epidemiological research on upper respiratory tract infections necessitates the utilization of diverse multiplex PCR methods to comprehensively understand the dynamics of pathogen spread and infection patterns. This underscores the pivotal role of advanced diagnostic technologies in shaping effective public health strategies and interventions.

## Figures and Tables

**Figure 1 diagnostics-14-00734-f001:**
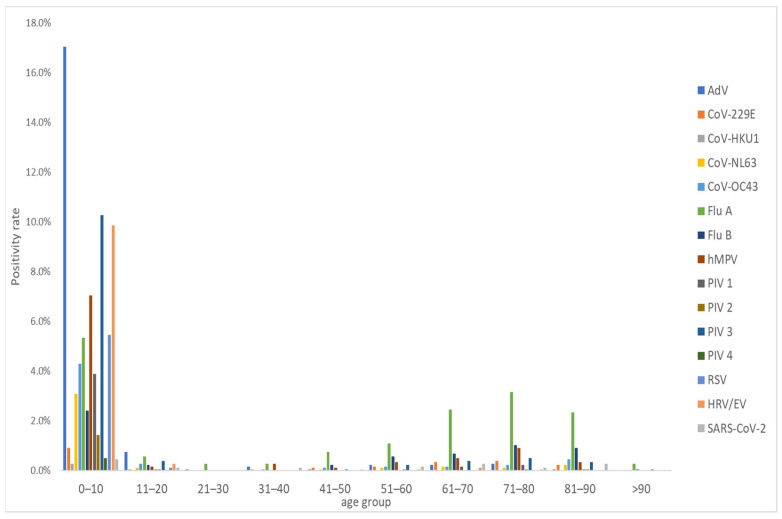
FilmArray™ Respiratory Panel (RP) positive detection of 1744 pathogens by age group collected over five years at Dankook University Hospital in Cheonan, Republic of Korea.

**Figure 2 diagnostics-14-00734-f002:**
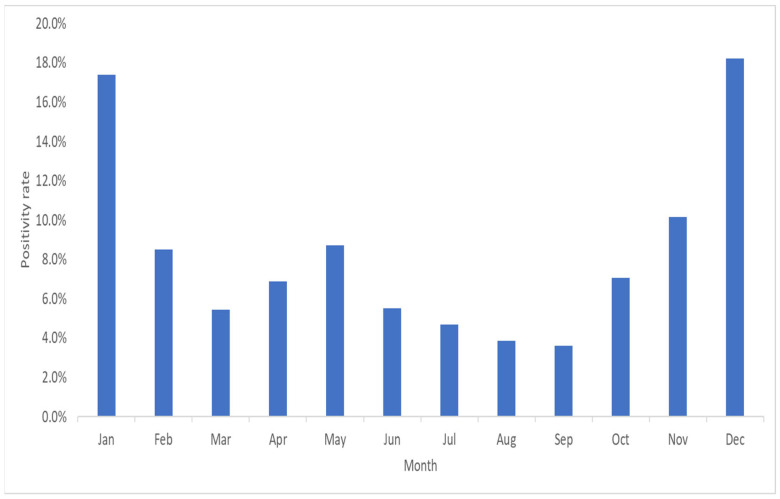
Monthly FilmArray™ Respiratory Panel (RP) positivity rate of 1744 pathogens detected over five years at Dankook University Hospital in Cheonan, Republic of Korea.

**Table 1 diagnostics-14-00734-t001:** Positivity rates of FilmArray™ Respiratory Panel pathogens.

Parameter	Positivity	
	Number of Samples	% of Total
All samples (*N* = 6367)		
Negative samples	4.829	75.8%
Positive samples	1538	24.1%
Single detections	1351	21.2%
Co-infections	187	2.9%
Co-infections (*n* = 187)		
Double infection	169	90.3%
Triple infection	17	9.1%
Quadruple infection	1	0.6%

**Table 2 diagnostics-14-00734-t002:** Detection and co-infection rates for each pathogen.

	Detected	Detection Rate (%)	Co-Infection	Co-Infection Rate (%)
AdV	329	18.9	93	28.3
CoV-229E	40	2.3	16	40.0
CoV-HKU1	5	0.3	3	60.0
CoV-NL63	68	3.9	26	38.2
CoV-OC43	101	5.8	31	30.7
SARS-CoV-2	28	1.6	6	21.4
hMPV	170	9.7	29	17.1
HRV/EV	181	10.4	57	31.5
FluA	289	16.6	38	13.1
FluB	107	6.1	15	14.0
PIV1	77	4.4	15	19.5
PIV2	29	1.7	4	13.8
PIV3	214	12.3	29	13.6
PIV4	9	0.5	5	55.6
RSV	97	5.6	26	26.8
Total	1744	100.0	393	22.5

AdV, adenovirus; CoV-229E, coronavirus 229E; CoV-HKU1, coronavirus HKU1; CoV-NL63, coronavirus NL63; CoV-OC43, coronavirus OC43; SARS-CoV-2, severe acute respiratory syndrome coronavirus 2; hMPV, human metapneumovirus; HRV/EV, human rhinovirus/enterovirus; FluA, influenza virus A; FluB, influenza virus B; PIV1, parainfluenza virus 1; PIV2, parainfluenza virus 2; PIV3, parainfluenza virus 3; PIV4, parainfluenza virus 4; RSV, respiratory syncytial virus.

**Table 3 diagnostics-14-00734-t003:** Distribution of co-infections of respiratory pathogens.

Single Infection		Double Infection			Triple Infection		
AdV	236 (17.5)	AdV	CoV-229E	2 (1.2)	AdV & CoV-NL63	CoV-229E	1 (5.9)
CoV-229E	24 (1.8)		CoV-OL63	4 (2.4)		CoV-OC43	1 (5.9)
CoV-HKU1	2 (0.1)		CoV-OC43	11 (6.5)		PIV3	1 (5.9)
CoV-NL63	42 (3.1)		hMPV	13 (7.7)		HRV/EV	1 (5.9)
CoV-OC43	70 (5.2)		HRV/EV	14 (8.3)	AdV & HRV/EV	PIV1	1 (5.9)
SARS-CoV-2	22 (1.6)		FluA	10 (5.9)		RSV	2 (11.8)
HRV/EV	141 (10.4)		FluB	2 (1.2)		CoV-OC43	1 (5.9)
RV/EV	124 (9.2)		PIV1	2 (1.2)		PIV3	3 (17.6)
FluA	251 (18.6)		PIV2	1 (0.6)	AdV & CoV-OC43	CoV-HKU1	1 (5.9)
FluB	92 (6.8)		PIV3	13 (7.7)		PIV3	1 (5.9)
PIV1	62 (4.6)		PIV4	1 (0.6)	AdV & RSV	SARS-CoV-2	1 (5.9)
PIV2	25 (1.9)		RSV	5 (3.0)	CoV-NL63 & PIV1	PIV3	1 (5.9)
PIV3	185 (13.7)	CoV-229E	CoV-OL63	2 (1.2)	CoV-NL63 & HRV/EV	RSV	1 (5.9)
PIV4	4 (0.3)		FluA	8 (4.7)	CoV-OC43 & hMPV	FluA	1 (5.9)
RSV	71 (5.3)		FluB	3 (1.8)		total	17 (100)
Total	1351 (100)	CoV-HKU1	hMPV	1 (0.6)			
			FluA	1 (0.6)			
		CoV-OL63	CoV-OC43	2 (1.2)			
			hMPV	2 (1.2)			
			HRV/EV	1 (0.6)			
			FluA	4 (2.4)			
			FluB	1 (0.6)			
			PIV1	1 (0.6)			
			PIV3	2 (1.2)			
		C.OC43	HRV/EV	1 (0.6)			
			FluA	4 (2.4)			
			FluB	3 (1.8)			
			PIV3	3 (1.8)			
			RSV	1 (0.6)			
		SARS	hMPV	1 (0.6)			
			HRV/EV	2 (1.2)			
			FluA	1 (0.6)			
			RSV	1 (0.6)			
		MV	HRV/EV	4 (2.4)			
			FluB	1 (0.6)			
			PIV1	2 (1.2)			
			PIV2	1 (0.6)			
			PIV3	3 (1.8)			
		HRV/EV	FluA	1 (0.6)			
			PIV1	5 (3.0)			
			PIV2	1 (0.6)			
			PIV3	4 (2.4)			
			RSV	14 (8.3)			
		FluA	FluB	5 (3.0)			
			PIV1	1 (0.6)			
			PIV2	1 (0.6)			
			PIV4	1 (0.6)			
		PIV1	PIV3	1 (0.6)			
			RSV	1 (0.6)			
			Total	169 (100)			

AdV, adenovirus; CoV-229E, coronavirus 229E; CoV-HKU1, coronavirus HKU1; CoV-NL63, coronavirus NL63; CoV-OC43, coronavirus OC43; SARS-CoV-2, severe acute respiratory syndrome coronavirus 2; hMPV, human metapneumovirus; HRV/EV, human rhinovirus/enterovirus; FluA, influenza virus A; FluB, influenza virus B; PIV1, parainfluenza virus 1; PIV2, parainfluenza virus 2; PIV3, parainfluenza virus 3; PIV4, parainfluenza virus 4; RSV, respiratory syncytial virus.

## Data Availability

The datasets used and analyzed during this current study are available from the corresponding author upon reasonable request.

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
