# Peer review of "Epidemiological Characterization of Respiratory Pathogens Using the Multiplex PCR FilmArray™ Respiratory Panel"

_diagnostics, 2024, doi:10.3390/diagnostics14070734_

Round 1
Reviewer 1 Report
Comments and Suggestions for Authors
The study is very interesting for understanding respiratory epidemiology. In the 5 years of the study, the COVID pandemic years were included. The panel previously did not include COVID. It would be interesting to see when specifically COVID detection was included in the panel and if during the pandemic time other types of co-infections were detected.
It is also said that virus detection was lower in summer. It would be interesting to say which microorganisms are more prevalent in that period and to what it may be due.
It would also be interesting and advisable to discuss about the detection of 3 or 4 microorganisms, since it may be due to false positives instead of coinfections, and if they have done other types of checks by other molecular biology teams in these cases.
Reviewer 2 Report
Comments and Suggestions for Authors
The current study titled “Epidemiological Characterization of Respiratory Pathogens Using the Multiplex PCR FilmArray™ Respiratory Panel” Ref: 2899970, deals with an important subject. It describes the importance of the titled technique for accurate detection of viral/bacterial infections due to upper respiratory tract infection. This is so essential for the appropriate treatment for infected patients in appropriate time for rapid recovery and safe human health. However minor points may be reconsidered by the author.
- Although “treatment planning” was mentioned as a keyword for this study, no considerable description was noticed in this study.
- No comparison was noticed by the titled technique and one of the most usable/successful traditional techniques mainly due to the efficacy, and time needed for final observed results in addition to expenses for each.
- Limitation of the study for narrow area for collecting testing samples should also be taken into consideration (may be in a future study).
Round 2
Reviewer 1 Report
Comments and Suggestions for Authors
Given that the authors do not have the option of having checked by another technique the positivity of the samples, it would be interesting and advisable to discuss the detection of 3 or 4 microorganisms, since it may be due to false positives instead of coinfections, being a limitation of the technique. And in this regard see literature supporting the detection of multiple microorganisms by PCR and its justification.
